# UNDERSTANDING DEEP LEARNING GENERALIZATION BY MAXIMUM ENTROPY

## ABSTRACT

Deep learning achieves remarkable generalization capability with overwhelming number of model parameters. Theoretical understanding of deep learning generalization receives recent attention yet remains not fully explored. This paper attempts to provide an alternative understanding from the perspective of maximum entropy. We first derive two feature conditions that softmax regression strictly apply maximum entropy principle. DNN is then regarded as approximating the feature conditions with multilayer feature learning, and proved to be a recursive solution towards maximum entropy principle. The connection between DNN and maximum entropy well explains why typical designs such as shortcut and regularization improves model generalization, and provides instructions for future model development.

## 1 INTRODUCTION

Deep learning has achieved significant success in various application areas. Its success has been widely ascribed to the remarkable generalization ability. Recent study shows that with very limited training data, a 12-layer fully connected neural network still generalizes well while kernel ridge regression easily overfits with polynomial kernels of more than 6 orders (Wu et al., 2017). Classical statistical learning theories like Vapnik-Chervonenkis (VC) dimension (Maass, 1994) and Rademacher complexity (Neyshabur et al., 2015) evaluate generalization based on the complexity of the target function class. It is suggested that the models with good generalization capability are expected to have low function complexity. However, most successful deep neural networks already have over 100 hidden layers, e.g., ResNet (He et al., 2016) and DenseNet (Huang et al., 2016) for image recognition. The number of model parameters in these cases is even larger than the number of training samples. Statistical learning theory cannot well explain the generalization capability of deep learning models (Zhang et al., 2017).

Maximum Entropy (ME) is a general principle for designing machine learning models. Models fulfilling the principle of ME make least hypothesis beyond the stated prior data, and thus lead to least biased estimate possible on the given information (Jaynes, 1957). Appropriate feature functions are critical in applying ME principle and largely decide the model generalization capability (Berger et al., 1996). Different selections of feature functions lead to different instantiations of maximum entropy models (Malouf, 2002; Yusuke & Jun'ichi, 2002). The most simple and well-known instantiation is that ME principle invents identical formulation of softmax regression by selecting certain feature functions and treating data as conditionally independent (Manning & Klein, 2003). It is obvious that softmax regression has no guaranty of generalization, indicating that inappropriate feature functions and data hypothesis violates ME principle and undermines the model performance. It remains not fully studied how to select feature functions to maximally fulfill ME principle and guarantee the generalization capability of ME models. Maximum entropy provides a potential but not-ready way to understand deep learning generalization.

This paper is motivated to improve the theory behind applying ME principle and use it to understand deep learning generalization. We research on the feature conditions to equivalently apply ME principle, and indicates that deep neural networks (DNN) is essentially a recursive solution to approximate the feature conditions and thus maximally fulfill ME principle.

- In Section 2, we first revisit the relation between generalization and ME principle, and conclude that models well fulfilling ME principle requires least data hypothesis so to possess good generalization capability. One general guideline for feature function selection is to transfer the hypothesis on input data to the constrain on model features [1]. This demonstrates the role of feature learning in designing ME models.

- Section 3 addresses what features to learn. Specifically, we derive two feature conditions to make softmax regression strictly equivalent to the original ME model (denoted as *Maximum Entropy Equivalence Theorem*). That is, if the utilized features meet the two conditions, simple softmax regression model can fulfill ME principle and guarantee generalization. These two conditions actually specify the goal of feature learning.

- Section 4 resolves how to meet the feature conditions and connects DNN with ME. Based on Maximum Entropy Equivalence Theorem, viewing the output supervision layer as softmax regression, the DNN hidden layers before the output layer can be regarded as learning features to meet the feature conditions. Since the feature conditions are difficult to be directly satisfied, they are optimized and recursively decomposed to a sequence of manageable problems. It is proved that, standard DNN uses the composition of multilayer non-linear functions to realize the recursive decomposition and uses back propagation to solve the corresponding optimization problem.

- Section 5 employs the above ME interpretation to explain some generalization-related observations of DNN. Specifically, from the perspective of ME, we provide an alternative way to understand the connection between deep learning and Information Bottleneck (Shwartz-Ziv & Tishby, 2017). Theoretical explanations on typical generalization design of DNN, e.g., shortcut, regularization, are also provided at last.

The contributions are summarized in three-fold:

1. We derive the feature conditions that softmax regression strictly apply maximum entropy principle. This helps understanding the relation between generalization and ME models, and provides theoretical guidelines for feature learning in these models.

2. We introduce a recursive decomposition solution for applying ME principle. It is proved that DNN maximally fulfills maximum entropy principle by multilayer feature learning and softmax regression, which guarantees the model generalization performance.

3. Based on the ME understanding of DNN, we provide explanations to the information bottleneck phenomenon in DNN and typical DNN designs for generalization improvement.

## 2 Revisiting Generalization and Maximum Entropy

In machine learning, one common task is to fit a model to a set of training data. If the derived model makes reliable predictions on unseen testing data, we think the model has good generalization capability. Traditionally, overfitting refers to a model that fits the training data too well but generalize poor to testing data, while underfitting refers to a model that can neither fits the training data nor generalize to testing data (Vapnik & Vapnik, 1998).

As a criterion for learning machine learning models, ME principle makes null hypothesis beyond the stated prior data $(X, Y)$ where $X, Y$ denote the original sample representation and label respectively. To facilitate the discussion between generalization and maximum entropy, we revisit generalization, overfitting and underfitting by how much data hypothesis is assumed by the model:

- **Underfitting**: Underfitting occurs when the model's data hypothesis is not satisfied by the training data.

- **Overfitting**: Overfitting occurs when the model's data hypothesis is satisfied by the training data, but not satisfied by the testing data.

---

[1] Here "feature" refers to the variable directly used by machine learning models like softmax regression. To avoid confusion, from now on, we will refer to the feature function in ME models as "predicate function" (Jeon & Manmatha, 2004).

- **Generalization**: According to ME principle, a model with good generalization capability is expected to have as less extra hypothesis on data $(X, Y)$ as possible.

The above interpretation of underfitting and overfitting can be illustrated with the toy example in Fig. 1(left). The underfitting model in solid line assumes linear relation on $(X, Y)$, which is not satisfied by the training data. The model in dot dash line assumes 5-order polynomial relation on $(X, Y)$, which perfectly fits to the training data. However, it is obvious that the hypothesis generalizes poorly to testing data and the 5-order polynomial model tends to overfitting. A coarse conclusion reaches that, introducing extra data hypothesis, whether or not fitting well to the training data, will lead to degradation of model generalization capability.

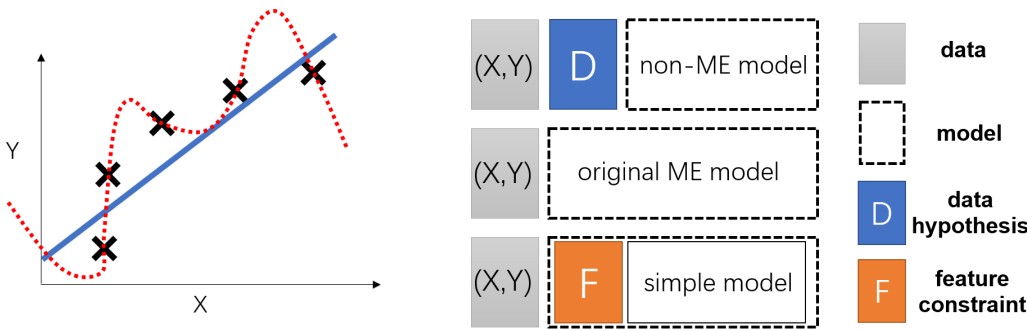

Figure 1: (**Left**) fitting 5 data points with linear (solid line) and 5-order (dot dash line) polynomials. (**Right**) illustration of different model settings with/without data hypothesis; from top to bottom: non-ME model with data hypothesis, original ME model without data hypothesis, simple model with feature constraint (equivalent to original ME).

One question arises: why ME models cannot guarantee good generalization? Continuing the discussion in *Introduction*, to enable the enumeration of predicate states, most ME models explicitly or implicitly introduce extra data hypothesis, e.g., softmax regression assumes independent observations when applying ME principle. Imposing extra data hypothesis actually violates the ME principle and degrades the model to non-ME (Maximum Entropy) model. The dilemma is: generalization requires no extra data hypothesis, but it is difficult to derive simple models without data hypothesis. Is there solution to apply ME principle without imposing hypothesis on the original data?

While the input original data $(X, Y)$ is fixed and maybe not compatible with the hypothesis, we can introduce model feature $T$ sufficient to represent data, and transfer the data hypothesis to feature constraint. Ideally the model defined on feature $T$ is a simple ME model (e.g., softmax regression), so that we can easily apply ME principle without imposing extra data hypothesis. In this case, the simple model plus feature constraint constitutes an equivalent implementation to ME principle and possesses good generalization capability. Fig. 1(right) illustrates these model settings with/without data hypothesis. It is easy to see that, from the perspective of applying ME, feature learning works between data and feature, with goal to realizing the feature constraints.

## 3 FEATURE CONDITIONS FOR MAXIMUM ENTROPY EQUIVALENCE

According to the above discussions, when applying ME principle, the problem becomes how to identify the equivalent feature constraints and simple models. Since the output layer of DNN is usually softmax regression, this section will explore what feature constraints can make softmax regression equivalent to the original ME model.

### 3.1 REVIEW OF ORIGINAL ME MODEL & FEATURE-BASED SOFTMAX MODEL

We first review the definition of original ME model and feature-based softmax model in this subsection. Note that in defining the original ME model, instead of using predicate functions as most ME

models, we deliver the constraints of $(X, Y)$ with joint distribution equality [2]. Before defining the softmax model, to facilitate the transfer of data hypothesis to feature constraint, we first provide the definition of feature $T$ over input data $X$, and then derive the general formulation of feature-based ME model. Feature-based softmax model can be seen as a special case of feature-based ME model.

**Definition 1.** *(Original Maximum Entropy Model)   Supposing the dataset has input $X$ and label $Y$, the task is to find a good prediction of $Y$ using $X$. The prediction $\hat{Y}$ needs to maximize the conditional entropy $H(\hat{Y}|X)$ while preserving the same distribution with data $(X, Y)$. This is formulated as:*

$$\min -H(\hat{Y}|X) \tag{1}$$

$$s.t. \ P(X, Y) = P(X, \hat{Y}), \ \sum_{\hat{Y}} P(\hat{Y}|X) = 1$$

*This optimization question can be solve by lagrangian multiplier method :*

$$\mathcal{L} = \sum_{X, \hat{Y}} P(X)P(\hat{Y}|X)log(P(\hat{Y}|X)) + \omega_0 \left(1 - \sum_{\hat{Y}} P(\hat{Y}|X)\right)$$
$$+ \sum_{X, Y} \omega_i \left(P(X, Y) - P(X)P(\hat{Y} = Y|X)\right)$$

*The above equation can be equivalently written with the original defined predicate function in (Berger et al., 1996):*

$$\mathcal{L} = \sum_{X, \hat{Y}} P(X)P(\hat{Y}|X)log(P(\hat{Y}|X)) + \omega_0 \left(1 - \sum_{\hat{Y}} P(\hat{Y}|X)\right)$$
$$+ \sum_{i} \omega_i \left(\sum_{X, Y} P(X, Y)f_i(X, Y) - \sum_{X, \hat{Y}} P(X)P(\hat{Y}|X)f_i(X, \hat{Y})\right)$$

*where $f_i(X, Y)$ is predicate function, which equalizes 1 when $(X, Y)$ satisfies a certain status:*

$$f_i(X, Y) = \begin{cases} 1, & X = x_i, Y = y_i \\ 0, & others \end{cases} \tag{2}$$

*The solution to the above problem is:*

$$P_\omega(\hat{Y} = y|X = x) = \frac{1}{Z_\omega(x)} exp \left(\sum_i \omega_i f_i(x, y)\right) \tag{3}$$

$$Z_\omega(x) = \sum_y exp \left(\sum_i \omega_i f_i(x, y)\right) \tag{4}$$

**Definition 2.** *(Feature-based Maximum Entropy Model)   $T$ is defined as a set of features over input $X$: $T$ is only related to $X$, and $t_i$ denotes the confidence of feature $T_i$ happening in input status $x$. In other words, $t_i(x) = P(T_i = 1|X = x)$, and $P(T_i = 0|X = x) = 1 - t_i(x)$.*

*According to the above definition of feature $T$, feature-based maximum entropy model can be formulated as :*

$$\min -H(\hat{Y}|T) \tag{5}$$

$$s.t. \ E_{P(T, Y)} = E_{P(T, \hat{Y})}, \ \sum_{\hat{Y}} P(\hat{Y}|T) = 1$$

---

[2] In the early paper introducing ME models, predicate functions are used to instantiate ME principle for model derivation. It is easily to find that according to the original predicate function defined in (Berger et al., 1996), the expectation over predicate functions is exactly joint distribution.

**Definition 3.** *(Feature-based Softmax Model)* *Feature-based softmax model is a special case of feature-based ME model with manageable predicate functions. The loglinear solution of feature-based softmax regression model based on feature $T$ is:*

$$P(\hat{Y} = y|X = x) = \frac{1}{Z(x)} exp\left( b(y) + \sum_i \lambda_i(y)t_i(x) \right) \qquad (6)$$

$$Z(x) = \sum_y exp\left( b(y) + \sum_i \lambda_i(y)t_i(x) \right) \qquad (7)$$

*where $\lambda_i(y)$ and $b(y)$ denote functions of $y$: $\lambda_i(y)$ is weight for feature $T_i$, and $b(y)$ is the bias term in softmax regression.*

### 3.2 MAXIMUM ENTROPY EQUIVALENCE THEOREM

From Eqn. (3) and Eqn. (4), we find it impossible to traverse all status of $(X, Y)$, making the original ME problem difficult to solve. To address this, many studies are devoted to designing special kind of predicate functions to make the problem solvable. However, recalling the discussion on ME and generalization in Section 2, if extra data hypothesis is imposed on $(X, Y)$, the generalization capability of the derived ME model will be undermined. An alternative solution is to design the predicate function by imposing constraints on intermediate feature $T$ instead of directly on input data $(X, Y)$.

On imposing the feature constraints, two issues need to be considered: (1) not arbitrary $T$ makes the feature-based ME model equivalent to the original ME model; (2) under the premise of equivalence, $T$ should make the derived ME model solvable (like the softmax regression). Based on these considerations, we prove and derive two necessary and sufficient feature conditions to make feature-based softmax regression (*Definition 3*) strictly equivalent to the original ME model (*Definition 1*). The theorem is denoted as *Maximum Entropy Equivalence Theorem*.

**Theorem** (Maximum Entropy Equivalence Theorem). *Given the input data $X, Y$ and feature $T$, the necessary and sufficient conditions of feature-based softmax model equivalent to the original maximum entropy model are:*

*<**condition 1**>: $X$ and $Y$ are conditionally independent given $T$;*
*<**condition 2**>: all features of $T$: $\{T_1, \cdots, T_i, \cdots, T_n\}$ are conditionally independent given $Y$.*

The proof to the theorem is given in Section A in the Appendix. The first condition ensures that feature-based ME model is equivalent to the original ME model, and thus be denoted as *equivalent condition*. The second condition makes feature-based ME model solvable and converted as feature-based softmax regression problem. We denote the second condition as *solvable condition*. This theorem on one hand derives operable feature constraints that softmax regression is equivalent to the original ME model, on the other hand provides theoretical guidance to feature learning with goal of improving model generalization.

## 4 MODELING DNN AS MAXIMUM ENTROPY

Based on the derived *Maximum Entropy Equivalence Theorem*, the original ME model is equivalent to a feature-based softmax model with two feature constraints. In this way, from the perspective of maximum entropy, if DNN uses softmax as the output layer, the previous latent layers can be seen as the process of feature learning to approach these constraints. However, these feature constraints are difficult to be satisfied directly, and therefore being decomposed to many smaller and manageable problems for approximation. This section claims that DNN actually uses the composition of multilayer non-linear functions to realize a recursive decomposition towards these feature constraints. In the following we will first introduce the recursive decomposition operation to difficult problem, and then prove that DNN with sigmoid-activated hidden layers and softmax output layer is exactly a recursive decomposition solution towards the original ME model.

### 4.1 RECURSIVE DECOMPOSITION

A common way to solve a difficult problem is relaxing the problem to an easier one, like majorize-minimize algorithms (Hunter & Lange, 2004). Inspired by this, we introduce a special case of relaxation solution to difficult problem: recursive decomposition. Decomposable problem is first defined as follows:

**Definition 4.** *(Decomposable problem) If a difficult optimization problem is equivalent to a manageable problem, but with additional constraints only related to extra added parameters, this problem is a decomposable problem.*

Obviously, according to *Maximum Entropy Equivalence Theorem*, the original ME problem is such a decomposable problem. If the original problem $P$ is decomposable, and $P$ is equivalent to a manageable problem $P_1$ with additional constraints $C_1$, we denote it as $P = P_1 + C_1$. In this case, we can solve $P_1 + C_1$ instead of directly solving $P$.

Since $P_1$ is easy to solve, it remains to satisfy the constraint $C_1$. The constrain $C_1$ can be approximately satisfied by an optimization problem $p_1$ as its upper bound. From *Definition 4*, we know that $p_1$ is only related to the extra added parameters. Now, we have $P = P_1 + p_1$.

If $p_1$ is solvable, we can use an algorithm similar to EM to solve $P_1 + p_1$:

**(1)** fix parameters in $p_1$ and optimize $P_1$;

**(2)** fix parameters in $P_1$ and optimize $p_1$;

**(3)** iterate (1) and (2) until convergence.

However, sometimes $p_1$ is still difficult to solve but decomposable. In this case, we need further decompose $p_1$ to a manageable problem $P_2$ with smaller problem $p_2$ under condition that $p_1 = P_2 + p_2$. The problem transfers to solve $P = P_1 + P_2 + p_2$ in a similar iterative way. If $p_2$ is still difficult, we can repeat this process to get $p_3, p_4, \cdots, p_L$ until $p_L$ is small enough that $p_L \approx P_L$ and $P_L$ is manageable. Since this constitutes a recursive process, we denote this way of relaxation as recursive decomposition.

The optimization process of recursive decomposition is also recursive. Given the decomposition of difficult problem $P = P_1 + \cdots + P_l + + P_L$, we have the following optimization process:

**(1)** fix parameters in $P_2, \cdots, P_L$ and optimize $P_1$;

**(2)** fix parameters in $P_1, P_3, \cdots, P_L$ and optimize $P_2$;

 $\cdots$

**(L)** fix parameters in $P_1, P_2, \cdots, P_{L-1}$ and optimize $P_L$;

**(L+1)** iterate (1), (2), $\cdots$, (L) until convergence.

The premise behind this method is that, if we change the constraints of problem to a minimum problem of its upper bound, the new problem is still a better approximation than the original problem without constraint.

### 4.2 DNN IS RECURSIVE DECOMPOSITION SOLUTION TOWARDS ME

This subsection will explain that DNN is actually a recursive decomposition solution towards maximum entropy, and the back propagation algorithm is a realization of parameter optimization to the model. According to *Maximum Entropy Equivalence Theorem*, the original ME model is equivalent to softmax model with two feature constraints, which is a typical decomposable problem. In the following we employ the above introduced recursive decomposition method to solve it: the original ME problem is the difficult problem $P$, softmax model is the manageable problem $P_1$, and the two conditions constitutes the constraints $C_1$ related only to feature $T$.

While the feature constraints $C_1$ are still difficult to be satisfied, we relax the constraints to smaller problems using the following *Feature Constraint Relaxation Theorem*.

**Theorem** (Feature Constraint Relaxation Theorem). *The constraints in Maximum Entropy Equivalence Theorem on feature $T = T_1, T_2, \cdots, T_n$:*

*(1) mutual information $I(X;Y|T) = 0$*

*(2) conditional mutual information $I(T_i, T_j|Y) = 0$ , $for\ all\ i \neq j$*

*can be relaxed to the following optimization problem:*

$$\min_T -\sum_i \lambda_i H(Ti|X) \tag{8}$$
$$s.t.\ E_{P(X,Y)} = E_{P(X,S(T))}$$

*where $S(T)$ denotes the output of softmax model if input is $T$.*

This theorem is proved in Section B in the Appendix. The above relaxed minimization problem constitutes $p_1$, which optimizes feature $T = T_1, T_2, ..., T_n$. Using the derivation from the proof for the above theorem, we know that minimization of $-\sum_i \lambda_i H(T_i|X)$ leads to $I(T_i; T_j|Y) = 0\ for\ all\ i \neq j$. The fact that $T_i, T_j$ is independent allows to split $p_1$ further to $n$ smaller problems $p_{11}, \cdots, p_{1i}, \cdots, p_{1n}$, where $p_{1i}$ is an optimization problem with the same formulation as Eqn. (8) but defined over $T_i$.

Note that the new optimization problems $p_{1i}$ are still difficult ME problems, which need to be decomposed and relaxed recursively till problem $p_{Li} \approx P_{Li}$ where $P_{Li}$ is manageable. According to *Maximum Entropy Equivalence Theorem*, each decomposed manageable problem $P_{li}$ is realized by a softmax regression. Since feature $T_i$ is binary random variable, the models for feature learning change to logistic regression. For a $L$-depth recursive decomposition, the original ME model is approximated by $\sum_{l=1}^L n_l$ logistic regression models and one softmax regression model ($n_l$ denotes the number of features at the $l$-th recursion depth). It is easy to find that this structure perfectly matches a basic DNN model: the depth of recursion corresponds to the network hidden layer (but in opposite index, i.e., the $L$-th depth recursion corresponds to the 1st hidden layer), the number of features at each recursion correspond to the number of hidden neurons at each layer, and the logistic regression corresponds to one layer of linear regression with sigmoid activation function.

Therefore, we reach a conclusion that DNN is a recursive decomposition solution towards maximum entropy. The generalization capability is thus guaranteed under the ME principle. This explains why DNN is designed as composition of multilayer non-linear functions. Moreover, the model learning technique, backpropagation, actually follows the same spirits as the optimization process in recursive decomposition for DNN parameter optimization.

## 5 EXPLAINING DNN VIA MAXIMUM ENTROPY

After modeling DNN as a recursive decomposition solution towards ME, in this section, we use the ME theory to explain some generalization-related phenomenon about DNN and provide interpretations on DNN structure design. Specifically, Section 5.1 explains why Information Bottleneck exists in DNN, and Section 5.2 explains why certain DNN structure design can improve generalization.

### 5.1 EXPLAINING INFORMATION BOTTLENECK

In the Information Bottleneck (IB) theory (Tishby et al., 1999), given data $(X, Y)$, the optimization target is to minimize mutual information $I(X;T)$ while $T$ is a sufficient statistic satisfying $I(T;Y) = I(X;Y)$. Shwartz-Ziv & Tishby (2017) designed an experiment about DNN and found that the intermediate feature $T$ of DNN meets the IB theory: maximize $I(T;Y)$ while minimizing $I(X;T)$.

Now, we prove that the output of constraint problem in ME model is sufficient to satisfy the Information Bottleneck theory. In other words, basic DNN model with softmax output fulfills IB theory.

**Corollary** (Corollary of ME's interpretation on Information Bottleneck). *The output of maximum entropy problem*

$$\min_T -\sum_i \lambda_i H(Ti|X)\ \ s.t.\ E_{P(X,Y)} = E_{P(X,S(T))}$$

*is sufficient condition to the IB optimization problem:*

$$\min_T I(X;T)\ \ s.t. I(T;Y) = I(X;Y)$$

The proof of this corollary is available in Section C in the Appendix. Since DNN is an approximation towards ME, this result explains why DNN tends to increase $I(T; Y)$ while reduce $I(X; T)$ and the Information Bottleneck phenomenon in DNN.

## 5.2 EXPLAINING DNN DESIGN FOR GENERALIZATION

DNN has some typical generalization designs, e.g., shortcut, regularization, etc. This subsection explains why these designs can improve model generalization capability.

Shortcut is widely used in many CNN framework. The traditional explanation is that shortcut makes information flow more convenient, so we can train deeper networks (He et al., 2016). But this cannot explain why shortcut contributes to a better performance. According to the above modeling of DNN as ME, CNN is a special kind of DNN where we use part of input $X$ at each layer to construct the model. The actual input of CNN is related to the size of corresponding convolution kernel, and receives only part of $X$ within its receptive field. Shortcut enriches different size of receptive fields and thus reserve more information from $X$ during problem decomposition in the recursion process.

The regularization in DNN can be seen as playing similar role as the feature conditions in *Maximum Entropy Equivalence Theorem*. Achille & Soatto (2017) demonstrated that the regularization design, like sgd, L2-Norm, dropout, is equal to minimizing the mutual information $I(X; T)$. Since we have proved that $I(X; T) \geq I(Ti; Tj) \geq I(Ti; Tj|Y)$ in Appendix C, these regularization designs thus help to minimize the upper bounds of $I(Ti; Tj|Y)$ and approximate the *solvable condition*.

The ME modeling of DNN also sheds some light on the role of network depth in generalization performance. Following the recursive decomposition discussion, it seems network with more layers leads to deeper recursion and thus closer approximation towards ME. However, it is noted that we are using relaxed optimization to replace the original constraints. Considering the continuous minimization of upper bound, simple DNN with too many hidden layers may not always guarantees the performance. We emphasize that for those CNNs with good architecture, more hidden layers bring richer receptive fields and less loss of information in $X$. In this case, increasing network depth will contribute to generalization improvement.

## 6 CONCLUSION AND FUTURE WORK

This paper regards DNN as a solution to recursively decomposing the original maximum entropy problem. From the perspective of maximum entropy, we ascribe the remarkable generalization capability of DNN to the introduction of least extra data hypothesis. The future work goes in two directions: (1) first efforts will be payed to identifying connections with other generalization theories and explaining more DNN observations like the role of ReLu activation and redundant features; (2) the second direction is to improve and exploit the new theory to provide instructions for future model development of traditional machine learning as well as deep learning methods.

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

## APPENDIX

## A    PROOF OF MAXIMUM ENTROPY EQUIVALENCE THEOREM

The two feature conditions can be separately proved. Firstly, we prove the necessity and sufficiency of condition 1 (*equivalent condition*) for equivalence of feature-based ME model and original ME model. Secondly, condition 2 (*solvable condition*) guarantees the solution of feature-based ME model in a manageable form (i.e., softmax regression).

To prove this theorem, we first prove the following three Lemmas.

**Lemma 1.** *If $T$ is a set of random variables only related to $X$, and $T$ satisfies condition 1, i.e., mutual information $I(X;Y|T) = 0$, then*

$$P(X,Y) = P(X,\hat{Y}) \Leftrightarrow E_{P(T,Y)} = E_{P(T,\hat{Y})}$$

*Proof.* Since $T$ is a set of random variables only related to $X$, it is obvious to have

$$P(X,Y) = P(X,\hat{Y}) \Rightarrow E_{P(T,Y)} = E_{P(T,\hat{Y})} \tag{9}$$

So the task leaves to prove

$$P(X,Y) = P(X,\hat{Y}) \Leftarrow E_{P(T,Y)} = E_{P(T,\hat{Y})}$$

Recall that $T$ is a set of random variables only related to $X$, then

$$P(X,Y) = P(X,\hat{Y}) \Leftrightarrow P(X,T,Y) = P(X,T,\hat{Y})$$

We further have $T$ satisfying *condition 1*:

$$I(X;Y|T) = 0 \Rightarrow P(X,Y|T) = P(X|T)P(Y|T)$$
$$\Rightarrow P(X,T,Y) = P(T)P(X,Y|T) = P(T)P(X|T)P(Y|T) = P(X|T)P(T,Y)$$

Similarly, $X \to T \to \hat{Y}$ is Marcov chain, hence we have:

$$I(X;\hat{Y}|T) = 0 \Rightarrow P(X,T,\hat{Y}) = P(X|T)P(T,\hat{Y})$$

$T$ is defined feature function on $X$, so $P(X|T)$ is a constant. We further have:

$$E_{P(T,Y)} = E_{P(T,\hat{Y})} \Rightarrow E_{P(X,T,Y)} = E_{P(X,T,\hat{Y})} \Leftrightarrow E_{P(X,Y)} = E_{P(X,\hat{Y})} \tag{10}$$

Note that $E_{P(X,Y)} = E_{P(X,\hat{Y})}$ indicates that the predicate functions satisfy Eqn. (2) in the definition of original ME model, and thus is equivalent to $P(X,Y) = P(X,\hat{Y})$.

With Eqn. (9) and Eqn. (10), we finally have:

$$E_{P(T,Y)} = E_{P(T,\hat{Y})} \Leftrightarrow E_{P(X,Y)} = E_{P(X,\hat{Y})} \tag{11}$$

$\square$

**Lemma 2.** *If $T$ is a set of random variables only related to $X$, and $P(X,Y) = P(X,\hat{Y}) \Leftrightarrow E_{P(T,Y)} = E_{P(T,\hat{Y})}$, then: $T$ satisfies condition 1.*

*Proof.* Since $T$ is a set of random variables only related to $X$, we have

$$E_{P(X,Y)} = E_{P(X,\hat{Y})} \Leftrightarrow E_{P(X,T,Y)} = E_{P(X,T,\hat{Y})} \tag{12}$$

$X \to T \to \hat{Y}$ is Marcov chain, we have:

$$I(X;\hat{Y}|T) = 0 \Rightarrow P(X,T,\hat{Y}) = P(X|T)P(T,\hat{Y})$$

Additionally ,

$$E_{P(T,Y)} = E_{P(T,\hat{Y})} \Leftrightarrow E_{P(X,Y)} = E_{P(X,\hat{Y})} \Leftrightarrow E_{P(X,T,Y)} = E_{P(X,T,\hat{Y})}$$

So we can derive:

$$P(X,T,Y) = P(X|T)P(T,Y) \Rightarrow I(X;Y|T) = 0$$

$\therefore T \; meets \; condition 1$ $\square$

**Lemma 3.** *If $T$ is a set of random variables only related to $X$ that satisfies condition 1, and $E_{P(T,Y)} = E_{P(T,\hat{Y})}$, then:*

$$\min -H(\hat{Y}|X) \Leftrightarrow \min -H(\hat{Y}|T)$$

*Proof.* $T$ is a set of random variables only related to $X$:

$$Y \to X \to T \; is \; Marcov \; chain \; \Rightarrow I(T;Y) \leqslant I(X;Y) \tag{13}$$

$T$ satisfies *condition 1*, so:

$$Y \to T \to X \; is \; Marcov \; chain \; \Rightarrow I(T;Y) \geqslant I(X;Y) \tag{14}$$

With Eqn. (13) and Eqn. (14), we have: $I(T;Y) = I(X;Y)$

Further using *Lemma1*, we can derive:

$$E_{P(T,Y)} = E_{P(T,\hat{Y})} \Leftrightarrow E_{P(X,Y)} = E_{P(X,\hat{Y})} \Leftrightarrow E_{P(X,T,Y)} = E_{P(X,T,\hat{Y})} \tag{15}$$

Therefore, we get

$$I(T;\hat{Y}) = I(X;\hat{Y}) \Rightarrow H(\hat{Y}|X) = H(\hat{Y}|X) \tag{16}$$

$\therefore \min -H(\hat{Y}|X) \Leftrightarrow \min -H(\hat{Y}|T)$ $\square$

**Theorem** (Maximum Entropy Equivalence Theorem). *Given the input data $X, Y$ and feature $T$, the necessary and sufficient conditions of feature-based softmax model equivalent to the original maximum entropy model are:*

*<**condition 1**>: $X$ and $Y$ are conditionally independent given $T$;*

*<**condition 2**>: all features of $T$: $\{T_1, \cdots, T_i, \cdots, T_n\}$ are conditionally independent given $Y$.*

*Proof.* With *Lemma1*, *Lemma2* and *Lemma3*, we derive that *condition 1* is necessary and sufficient for the equivalence of original ME model and the following feature-based ME model:

$$\min \; -H(\hat{Y}|T)$$

$$s.t. \; E_{P(T,Y)} = E_{P(T,\hat{Y})}, \; \sum_{\hat{Y}} P(\hat{Y}|T) = 1$$

The above optimization problem can be solved with the following solution:

$$P_\omega(\hat{Y} = y|T) = \frac{1}{Z_\omega} exp\left(\sum_i \omega_i f_i(T, y)\right) \tag{17}$$

$$Z_\omega = \sum_y exp\left(\sum_i \omega_i f_i(T, y)\right) \tag{18}$$

However, this solution is too complex to apply. With $n$ features $T = \{T_1, T_2, ..., T_n\}$ and $m$ different classes of $Y$, there will be $m * 2^n$ different $f_i(T, Y)$. *Condition 2* assumes the conditional independence among feature $(T_i, T_j)$, which derives that the joint distribution equation $P(T, Y) = P(T, \hat{Y})$ is equivalent to its marginal distribution version $P(T_i, Y) = P(T_i, \hat{Y})$, $i = 1...n$. In this case, there leaves only $2 * m * n$ different $f_i(T, Y)$.

According to definition, for each $T_i$, we have $P(T_i = 1|X = x) = t_i(x)$ and $P(T_i = 0|X = x) = 1 - t_i(x)$. Therefore, under *condition 2*, the predicate functions will be:

$$f_{i1}(T_i = 1, y) = \begin{cases} t_i(x), & X = x, Y = y \\ 0, & others \end{cases}$$

$$f_{i0}(T_i = 0, y) = \begin{cases} 1 - t_i(x), & X = x, Y = y \\ 0, & others \end{cases}$$

We then have:

$$\omega_i f_i(T_i, y) = \omega_{i0} f_{i0}(T_i = 0, y) + \omega_{i1} f_{i1}(T_i = 1, y) = \omega_{i0} + (\omega_{i1} - \omega_{i0}) t_i(x)$$

where $\omega$ denotes variable about $y$.

We further define $b(y) = \sum_i \omega_{i0}$ and $\lambda_i(y) = (\omega_{i1} - \omega_{i0})$, then the solution of Eqn. (17) and Eqn. (18) change to:

$$P(\hat{Y} = y|X = x) = \frac{1}{Z(x)} exp\left(b(y) + \sum_i \lambda_i(y) t_i(x)\right) \tag{19}$$

$$Z(x) = \sum_y exp\left(b(y) + \sum_i \lambda_i(y) t_i(x)\right) \tag{20}$$

This is the identical formulation to the general softmax regression model as in *Definition 3*. It also explains why we have bias term in the softmax model. Note that $t_i(x)$ need not to be in range $[0, 1]$ when we use the softmax model, as we can change $\lambda$ and $b$ to achieve translation and scaling. $\quad \square$

## B    PROOF OF FEATURE CONSTRAINT RELAXATION THEOREM

**Theorem** (Feature Constraint Relaxation Theorem).    *The constraints in Maximum Entropy Equivalence Theorem on feature $T = T_1, T_2, ..., T_n$:*

*(1) $I(X; Y|T) = 0$*

*(2) $I(T_i, T_j|Y) = 0$ , for all $i \neq j$*

*can be relaxed to the following optimization problem:*

$$\min_{T} \ -\sum_{i} \lambda_i H(Ti|X)$$
$$\text{s.t. } E_{P(X,Y)} = E_{P(X,S(T))} \tag{21}$$

*where $S(T)$ denotes the output of softmax model if input is $T$.*

*Proof.*    Since $T$ is only related to $X$, $T_i \to X \to T_j$ is Marcov chain, and

$$I(T_i; T_j) \leqslant I(T_i; X) \Rightarrow \sum_{i \neq j} \lambda_{i,j} I(T_i; T_j) \leqslant \sum_{i \neq j} \lambda_{i,j} I(T_i; X)$$

We can relax the minimization problem to minimize its upper bound instead, so

$$\min \sum_{i \neq j} \lambda_{i,j} I(T_i; T_j) \Leftarrow \min \sum_{i} \lambda_i I(T_i|X) \Leftarrow \min \sum_{i} \lambda_i (H(X) - H(T_i|X))$$

Additionally, we have $I(T_i; T_j|Y) \leqslant I(T_i; T_j)$, hence:

$$\min \sum_{i \neq j} \lambda_{i,j} I(T_i; T_j|Y) \Leftarrow \min \sum_{i \neq j} \lambda_{i,j} I(T_i; T_j)$$

Since $H(X)$ is constant, so

$$I(T_i; T_j|Y) = 0 \ for \ all \ i \neq j \ \Leftarrow \ \min \ -\sum_{i} \lambda_i H(T_i|X)$$

Note that $\hat{Y} = S(T)$:
$$X \to T \to \hat{Y} \ is \ Marcov \ chain$$

Recall that $\hat{Y}$ is solution to the problem $P_1$, and $P_1$ has constraint $E_{P(X,Y)} = E_{P(X,\hat{Y})}$. Same as $E_{P(X,Y)} = E_{P(X,S(T))}$, we have

$$X \to T \to \hat{Y} \to Y \ is \ Marcov \ chain$$
$$\Rightarrow \ I(X; Y|T) = 0$$

$\therefore$ the solution to the optimization problem in Eqn. (21) is sufficient to satisfy the constraints of $T$ in *Maximum Entropy Equivalence Theorem.*    □

## C    PROOF OF COROLLARY OF ME'S INTERPRETATION ON INFORMATION BOTTLENECK

**Lemma 4.**
$$\min \ -\sum_{i} \lambda_i H(Ti|X) \ \Rightarrow \ \min \ I(X; T)$$

*Proof.*

$$\min \ -\sum_{i} \lambda_i H(Ti|X) \ \Rightarrow \ \min \ -H(T|X) \ \Rightarrow \ \min \ I(X; T)$$

□

**Lemma 5.** *T is only related to X, then*

$$E_{P(X,Y)} = E_{P(X,S(T))} \Rightarrow I(T;Y) = I(X;Y)$$

*Proof.*

$$E_{P(X,Y)} = E_{P(X,S(T))} \Rightarrow Y \rightarrow T \rightarrow X \ is \ Marcov \ Chain \Rightarrow I(X;Y) \leqslant I(T;Y)$$

T is only related to X , so

$$Y \rightarrow X \rightarrow T \ is \ Marcov \ Chain \Rightarrow I(T;Y) \leqslant I(X;Y)$$

$$\therefore \ I(T;Y) = I(X;Y)$$

$\square$

**Corollary** (Corollary of ME's interpretation on Information Bottleneck)**.** *The output of maximum entropy problem*

$$\min_{T} \ -\sum_{i} \lambda_i H(Ti|X) \quad s.t. \ E_{P(X,Y)} = E_{P(X,S(T))}$$

*is sufficient condition to the IB optimization problem:*

$$\min_{T} \ I(X;T) \ s.t. I(T;Y) = I(X;Y)$$

*Proof.* Summing up *Lemma4* and *Lemma5*, the output of the constraint problem is sufficient to solving the IB optimization problem . $\square$

