# OpenReview forum: "Understanding Deep Learning Generalization by Maximum Entropy"
_ICLR.cc/2018/Conference — Reject_

### Official Review · AnonReviewer1 · 2017-11-29

**Rating:** 2
**Confidence:** 3

**Review:**

Summary:

This paper presents a derivation which links a DNN to recursive application of
maximum entropy model fitting. The mathematical notation is unclear, and in
one cases the lemmas are circular (i.e. two lemmas each assume the other is
correct for their proof). Additionally the main theorem requires complete
independence, but the second theorem provides pairwise independence, and the
two are not the same.

Major comments:

- The second condition of the maximum entropy equivalence theorem requires
  that all T are conditionally independent of Y. This statement is unclear, as
it could mean pairwise independence, or it could mean jointly independent
(i.e. for all pairs of non-overlapping subsets A & B of T I(T_A;T_B|Y) = 0).
This is the same as saying the mapping X->T is making each dimension of T
orthogonal, as otherwise it would introduce correlations. The proof of the
theorem assumes that pairwise independence induces joint independence and this
is not correct.

- Section 4.1 makes an analogy to EM, but gradient descent is not like this
  process as all the parameters are updated at once, and only optimised by a
single (noisy) step. The optimisation with respect to a single layer is
conditional on all the other layers remaining fixed, but the gradient
information is stale (as it knows about the previous step of the parameters in
the layer above). This means that gradient descent does all 1..L steps in
parallel, and this is different to the definition given.

- The proofs in Appendix C which are used for the statement I(T_i;T_j) >=
  I(T_i;T_j|Y) are incomplete, and in generate this statement is not true, so
requires proof.

- Lemma 1 appears to assume Lemma 2, and Lemma 2 appears to assume Lemma 1.
  Either these lemmas are circular or the derivations of both of them are
unclear.

- In Lemma 3 what is the minimum taken over for the left hand side? Elsewhere
  the minimum is taken over T, but T does not appear on the left hand side.
Explicit minimums help the reader to follow the logic, and implicit ones
should only be used when it is obvious what the minimum is over.

- In Lemma 5, what does "T is only related to X" mean? The proof states that
  Y -> T -> X forms a Markov chain, but this implies that T is a function of
Y, not X.

Minor comments:

- I assume that the E_{P(X,Y)} notation is the expectation of that probability
  distribution, but this notation is uncommon, and should be replaced with a
more explicit one.

- Markov is usually romanized with a "k" not a "c".

- The paper is missing numerous prepositions and articles, and contains
  multiple spelling mistakes & typos.

---

### Official Review · AnonReviewer3 · 2017-11-30
**extremely hard to follow, needs major revision**

**Rating:** 3
**Confidence:** 3

**Review:**

The paper aims to provide a view of deep learning from the perspective of maximum entropy principle.  I found the paper extremely hard to follow and seemingly incorrect in places.  Specifically:
a) In Section 2, the example given to illustrate underfitting and overfitting states that the 5-order polynomial obviously overfits the data.  However, without looking at the test data and ensuring the fact that it indeed was not generated by a 5-order polynomial, I don’t see how such a claim can be made.
b) In Section 2 the authors state “Imposing extra data hypothesis actually violates the ME principle and degrades the model to non-ME model.” … Statements like this need to be made much clearer, since imposing feature expectation constraints (such as Eq. (3) in Berger et al. 1996) is a perfectly legitimate construct in ME principle.
c) The opening paragraph of Section 3 is quite unclear; phrases like “how to identify the equivalent feature constraints and simple models” need to be made precise, it is not clear to me what authors mean by this.
d) I’m not able to really follow Definition 1, perhaps due to unclear notation.  It seems to state that we need to have P(X,Y) = P(X,\hat{Y}), and if that’s the case not clear what more can be accomplished by maximizing conditional entropy H(\hat{Y}|X).  Also, there is a spurious w_i in Definition 1.
e) Definition 2.  Not clear what is meant by notation E_{P(T,Y)}.
f) Definition 3 uses t_i(x) without defining those, and I think those are different from t_i(x) defined in Definition 2.

I think the paper needs to be substantially revised and clarified before it can be published at ICLR.

---

### Official Review · AnonReviewer2 · 2017-12-01
**This paper presented a theoretical result for the generalization of DNN using the maximum entropy principle.**

**Rating:** 6
**Confidence:** 2

**Review:**

The presentation of the paper is crisp and clear. The problem formulation is explained clearly and it is well motivated by theorems. It is a theoretical papers and there is no experimental section. This is the only drawback for the paper as the claims is not supported by any experimental section. The author could add some experiments to support the idea presented in the paper.

---

### Decision · Program_Chairs · 2018-01-29
**ICLR 2018 Conference Acceptance Decision**

**Decision:**

Reject

**Comment:**

The reviewers are in agreement, that the paper is a big hard to follow and incorrect in places, including some claims not supported by experiments.